# REVISITING INFORMATION-BASED CLUSTERING WITH PSEUDO-POSTERIOR MODELS

## ABSTRACT

Maximization of mutual information (MI) between the network's input and output motivates standard losses for unsupervised discriminative clustering enforcing "decisiveness" and "fairness". In the context of common softmax models, we clarify several general properties of such discriminative losses that were previously not well understood: the relation to K-means, or lack thereof, and *margin-maximization*. In particular, we show that "desiciveness" without the extra regularization term can lead to poor classification margins. Also, non-convexity of information-based losses motivates us to focus on self-supervised approaches introducing effective higher-order optimization algorithms with auxiliary variables. Addressing limitations of existing formulations, we propose a new self-supervised loss with soft auxiliary variables, or *pseudo-confidence* estimates. In particular, we introduce *strong* fairness and motivate the *reverse* cross-entropy as a robust loss for network training from noisy pseudo-confidence estimates. The latter is efficiently computed using variational inference - we derive a new EM algorithm with closed-form solutions for E and M steps. Empirically, our algorithm improves the performance of earlier methods for information-based clustering.

## 1 INTRODUCTION

We were inspired by the work of Bridle, Heading, and MacKay from 1991 Bridle et al. (1991) formulating *mutual information* (MI) loss for unsupervised discriminative training of neural networks using probability-type outputs, e.g. *softmax* $\sigma : \mathcal{R}^K \to \Delta^K$ mapping $K$ logits $l_k \in \mathcal{R}$ to a point in the probability simplex $\Delta^K$. Such output $\sigma = (\sigma_1, \ldots, \sigma_K)$ is often interpreted as a *pseudo posterior*[1] over $K$ classes, where $\sigma_k = \frac{\exp l_k}{\sum_i \exp l_i}$ is a scalar prediction for each class $k$.

The unsupervised loss proposed in Bridle et al. (1991) trains the model predictions to keep as much information about the input as possible. They derived an estimate of MI as the difference between the average entropy of the output and the entropy of the average output

$$L_{mi} \ := \ -MI(c, X) \ \approx \ \overline{H(\sigma)} \ - \ H(\overline{\sigma}) \tag{1}$$

where $c$ is a random variable representing class prediction, $X$ represents the input, and the averaging is done over all input samples $\{X_i\}_{i=1}^M$, *i.e.* over $M$ training examples. The derivation in Bridle et al. (1991) assumes that softmax represents the distribution $\Pr(c|X)$. However, since softmax is not a true posterior, the right hand side in (1) can be seen only as a *pseudo* MI loss. In any case, (1) has a clear discriminative interpretation that stands on its own: $H(\overline{\sigma})$ encourages "fair" predictions with a balanced support of all categories across the whole training data set, while $\overline{H(\sigma)}$ encourages confident or "decisive" prediction at each data point implying that decision boundaries are away from the training examples Grandvalet & Bengio (2004). Generally, we call clustering losses for softmax models "information-based" if they use measures from the information theory, e.g. entropy.

Discriminative clustering loss (1) can be applied to deep or shallow models. For clarity, this paper distinguishes parameters $\mathbf{w}$ of the *representation* layers of the network computing features $f_\mathbf{w}(X) \in \mathcal{R}^N$ for any input $X$ and the linear classifier parameters $\mathbf{v}$ of the output layer computing $K$-logit vector $\mathbf{v}^\top f$ for any feature $f \in \mathcal{R}^N$. The overall network model is defined as

$$\sigma(\mathbf{v}^\top f_\mathbf{w}(X)). \tag{2}$$

---

[1]"Pseudo" emphasizes that discriminative training does not lead to the true Bayesian posteriors, in general.

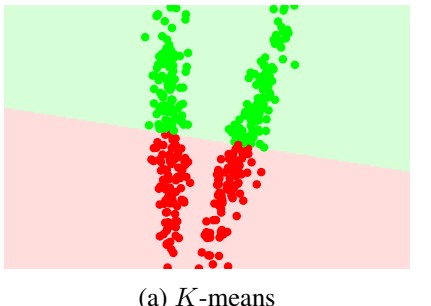 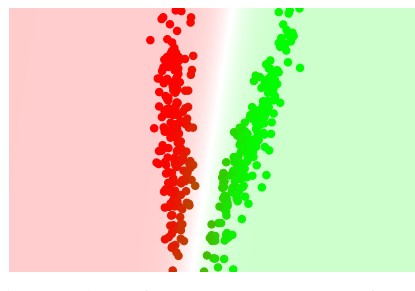

(a) $K$-means          (b) MI clustering (1,3) + max margin (7)

Figure 1: Generative vs Discriminative clustering - binary example ($K = 2$) for data points $X \in \mathcal{R}^N$ ($N = 2$) comparing linear methods of similar parametric complexity: (a) $K$-means $\mu_k \in \mathcal{R}^N$ and (b) MI clustering (1) using linear classifier (3) with $K$-column matrix $\mathbf{v}$ producing logits $\mathbf{v}_k^\top X$. Red and green colors show the optimal partitioning of the data points, as well as the corresponding decision regions over the whole 2D space. The linear decision boundary in (a) is "hard" since K-means outputs hard clustering $\arg\min_k \|x - \mu_k\|$. The "soft" margin in (b) is due to the *softmax* in the linear classifier (3) shown via the transparency channel $\alpha \propto \sigma_k$ for each region's color.

A special "shallow" case of the model in (2) is a basic linear discriminator

$$\sigma(\mathbf{v}^\top X) \qquad (3)$$

directly operating on low-level input features $f = X$. Optimization of the loss (1) for the shallow model (3) is done only over linear classifier parameters $\mathbf{v}$, but the deeper network model (2) is optimized over all network parameters $[\mathbf{v}, \mathbf{w}]$. Typically, this is done via gradient descent or backpropagation Rumelhart et al. (1986); Bridle et al. (1991).

Simple 2D example in Figure 1(b) motivates "decisiveness" and "fairness" as discriminative properties in the unsupervised clustering loss (1) in the context of a low-level linear classifier (3). MI clustering is compared with the standard $K$-means result in Figure 1(a). In this "shallow" 2D setting both clustering methods are linear and have similar parametric complexities, about $K \times N$ parameters. $K$-means (a) finds balanced compact clusters of the least squared deviations or variance. This can also be interpreted "generatively", see Kearns et al. (1997), as MLE-based fitting of two (isotropic) Gaussian densities, explaining the failure for non-isotropic clusters in (a). To fix (a) "generatively", one should use non-isotropic Gaussian densities, e.g. 2-mode GMM would produce soft clusters similar to (b). However, this has costly parametric complexity - two extra covariance matrices to estimate and quadratic decision boundaries. In contrast, there is no estimation of complex data density models in (b). MI loss (1) trains a simple linear classifier (3) to produce a balanced ("fair") decision boundary away from the data points ("decisiveness"). Later, we show that the "decisiveness" may be deficient without an extra *margin maximization* term, see Fig.2.

In the context of deep models (2), unsupervised MI loss finds non-linear partitioning of (unlabeled) training data points $\{X_i\}$ as the network learns their high-dimensional embedding $f_{\mathbf{w}}(X)$. In this case, the loss (1) is optimized with respect to both representation $\mathbf{w}$ and classification $\mathbf{v}$ parameters.

## 1.1 RELATED WORK ON DISCRIMINATIVE DEEP CLUSTERING

Unsupervised discriminative clustering via (pseudo) posterior models trained by MI loss Bridle et al. (1991) has growing impact Krause et al. (2010); Ghasedi Dizaji et al. (2017); Hu et al. (2017); Ji et al. (2019); Asano et al. (2020); Jabi et al. (2021) due to widespread of neural networks in computer vision where supervision is expensive or infeasible.

Clear information-theoretic and discriminative interpretations of MI clustering criterion (1) motivated many extensions. For example, Krause et al. (2010) proposed to reduce the complexity of the model by combining MI loss (1) with regularization of all network parameters interpreted as an isotropic Gaussian prior on these weights

$$
\begin{aligned}
L_{mi+decay} &= \overline{H(\sigma)} - H(\overline{\sigma}) + \|[\mathbf{v}, \mathbf{w}]\|^2 \\
&\stackrel{c}{=} \overline{H(\sigma)} + KL(\overline{\sigma} \,\|\, u) + \|[\mathbf{v}, \mathbf{w}]\|^2
\end{aligned} \qquad (4)
$$

where $\stackrel{c}{=}$ represents equality up to an additive constant and $u$ is a uniform distribution over $K$ classes. Of course, minimizing the norm of network weights as above corresponds to the *weight decay* - a common default for network training.

The second formulation of the loss (4) uses KL divergence motivated in Krause et al. (2010) by the possibility to generalize from "fairness" to balancing with respect to any given target distribution different from the uniform $u$. This paper keeps $u$ in all balancing KL terms, in part to preserve MI interpretation of the loss.

Empirically, network training with MI loss (4) can be improved using *self-augmentation* techniques Hu et al. (2017). Such techniques are common in deep clustering. Self-augmentation can also directly utilize MI. For example, Ji et al. (2019) maximize the mutual information between class predictions for input $X$ and its augmentation counterpart $X'$ encouraging high-level features invariant to augmentation.

Optimization of MI losses (1) or (4) during network training is mostly done with standard gradient descent or backpropagation Bridle et al. (1991); Krause et al. (2010); Hu et al. (2017). However, due to the entropy term representing the decisiveness, such loss functions are non-convex and present challenges to the gradient descent. This motivates alternative formulations and optimization approaches. For example, it is common to incorporate into the loss auxiliary variables $y$ representing *pseudo-labels* for unlabeled data points $X$ and to estimate them jointly with optimization of the network parameters Ghasedi Dizaji et al. (2017); Asano et al. (2020); Jabi et al. (2021). Typically, such *self-labeling* approaches to unsupervised network training iterate optimization of the loss over pseudo-labels and network parameters, similarly to the Lloyd's algorithm for $K$-means Bishop (2006). While the network parameters are still optimized via gradient descent, the pseudo-labels can be optimized via more powerful algorithms.

For example, Asano et al. (2020) formulate self-labeling using the following constrained optimization problem with discrete pseudo-labels $y$

$$L_{ce} \quad = \quad \overline{H(y,\sigma)} \qquad s.t. \quad y \in \Delta_{0,1}^K \quad and \quad \bar{y} = u \qquad (5)$$

where $\Delta_{0,1}^K$ are *one-hot* distributions, *i.e.* corners of the probability simplex $\Delta^K$. Training of the network predictions $\sigma$ is driven by the standard *cross entropy* loss $H(y,\sigma)$, which is convex assuming fixed (pseudo) labels $y$. With respect to variables $y$, the cross entropy is linear. Without the balancing constraint $\bar{y} = u$, the optimal $y$ corresponds to the hard $\arg\max(\sigma)$. However, the balancing constraint converts this into an integer programming problem that can be approximately solved via *optimal transport* Cuturi (2013). The cross-entropy in (5) encourages the network predictions $\sigma$ to approximate the estimated one-hot target distributions $y$, which implies the decisiveness.

Self-labeling methods for unsupervised clustering can also use soft pseudo-labels $y \in \Delta^K$ as target distributions in cross-entropy $H(y,\sigma)$. In general, soft targets $y$ are common in $H(y,\sigma)$, e.g. in the context of noisy labels Tanaka et al. (2018); Song et al. (2022). Softened targets $y$ can also assist network calibration Guo et al. (2017); Müller et al. (2019) and improve generalization by reducing over-confidence Pereyra et al. (2017). In the context of unsupervised clustering, cross entropy $H(y,\sigma)$ with soft pseudo-labels $y$ approximates the decisiveness since it encourages $\sigma \approx y$ implying $H(y,\sigma) \approx H(y) \approx H(\sigma)$ where the latter is the first term in (1). Instead of the hard constraint $\bar{y} = u$ used in (5), similarly to (4) the soft fairness constraint can be represented by KL divergence $KL(\bar{y} \,\|\, u)$, as in Ghasedi Dizaji et al. (2017); Jabi et al. (2021). In particular, Jabi et al. (2021) formulates the following self-supervised clustering loss

$$L_{ce+kl} \quad = \quad \overline{H(y,\sigma)} \quad + \quad KL(\bar{y} \,\|\, u) \qquad (6)$$

encouraging the decisiveness and fairness as discussed. Similarly to (5), the network parameters in loss (6) are trained by the standard cross-entropy term, but optimization over relaxed pseudo-labels $y \in \Delta^K$ is relatively easy due to convexity. While there is no closed form solution, the authors offer an efficient approximate solver for $y$. Iterating steps that estimate pseudo-labels $y$ and optimize the model parameters resembles the Lloyd's algorithm for K-means. The results in Jabi et al. (2021) also establish a formal relation between the loss (6) and the $K$-means objective.

## 1.2 MOTIVATION AND CONTRIBUTIONS

Our paper analyses several conceptual problems and limitations of the earlier work on information-based clustering with posterior models. Our goal is better understanding of the information-based

losses, which leads to their improved formulations and effective optimization algorithms for the non-convex information-based criteria. Similarly to some prior work, we focus on self-supervised approaches including auxiliary latent variables. We motivate novel formulations for the *decisiveness* and *fairness* constraints, argue for an explicit margin maximization term, and derive a new efficient EM algorithm for estimating auxiliary variables.

Our work is most closely related to Jabi et al. (2021) who proposed the self-supervised clustering loss (6) and the corresponding ADM algorithm. Their inspiring approach provides a good reference point for our proposal for a self-supervision loss (10). It also helps to illuminate problems with the standard formulations. In general, these problems are due to limited conceptual understanding of the information-based discriminative clustering losses.

In particular, we disagree with the main theoretical claim in Jabi et al. (2021) establishing a formal equivalence between K-means and regularized MI-based clustering. In fact, our Figure 1 works as a simple 2D counterexample to their claim[2]. Also, they extend the information-based loss with the classifier regularization $\|\mathbf{v}\|^2$, but this extra quadratic term is used mainly as a tool in their proof of algebraic similarity between their loss and the standard K-means loss[3]. We present a very different motivation for the regularization term. In contrast to prior work, we demonstrate that $\|\mathbf{v}\|^2$ is required in the discriminative information-based losses for explicit *margin maximization*.

Besides clarifying some useful general properties of the information-based losses for posterior models, we propose new technical ideas improving unsupervised and weakly-supervised training of such models. Below is a summary of the main contributions and their motivation:

• we propose the *forward* version of KL divergence as a *strong fairness* loss. The "strength" of the corresponding fairness or balancing constraint can be related to the well-known *zero-avoiding* property of the forward KL divergence. It is illustrated in Figure 3(a).

• we propose the *reverse* cross-entropy $H(\sigma, y)$ as a decisiveness constraint in the context of self-supervision. In general, this loss is non-standard for network training, in part because it is not well-defined for very common one-hot targets $y$. However, we show that there are important advantages in the context of self-labeling with soft auxiliary variables $y$. First, $H(\sigma, y)$ is a linear function for network predictions $\sigma$ such that, see Figure 3(b), the network training becomes robust to errors in $y$, which could be significant in self-labeling. The reversal makes the predictions ignore uncertain $y$ during training, rather than replicate their uncertainty. In fact, the reverse cross-entropy $H(\sigma, y)$ makes it difficult to continue calling distributions $y$ *soft labels* or *targets*. Instead, we will refer to the auxiliary variables $y \in \Delta^K$ as *pseudo-confidence* estimates. Note that the reverse cross entropy effectively turns predictions $\sigma$ into the strong targets for $y$. In combination with the *strong fairness*, the reverse cross entropy defines a well-constrained loss for estimating $y$.

• we show that regularization of the linear classifier $\|\mathbf{v}\|^2$ implies *margin-maximization* for the decision boundary that should be added to the decisiveness term to avoid its deficiency, see Figure 2

• a combination of reverse cross-entropy, strong fairness, and margin-maximization terms define our new self-supervised loss for clustering (10). It is convex w.r.t. $y$. We derive an efficient *expectation-maximization* (EM) algorithm for estimating $y$ with closed-form solutions in E and M steps.

• our empirical results show an improved performance w.r.t. the earlier information-based methods.

The rest of our paper is organized as follows. Section 2 motivates our information-based self-supervised clustering loss and derives our EM algorithm. Section 3 compares our approach with existing information-based discriminative clustering losses for unsupervised or self-supervised training of pseudo-posterior (softmax) models. Conclusions are provided in Section 4.

---

[2]The proof of Proposition 2 has a critical technical error - it ignores normalization for soft-max prediction in their equation (5), which is hidden via $\propto$ symbol. Such normalization is critical for pseudo-posterior models.

[3]Since they ignore normalization in the softmax prediction, then $\ln \sigma$ in the cross-entropy $H(y, \sigma)$ turns into a linear term w.r.t. logits $\mathbf{v}^\top x$. Adding regularization $\|\mathbf{v}\|^2$ to such loss allows them to create a quadratic form with respect to $\mathbf{v}$ that resembles squared errors loss in K-means, which is quadratic w.r.t means $\mu_k$).

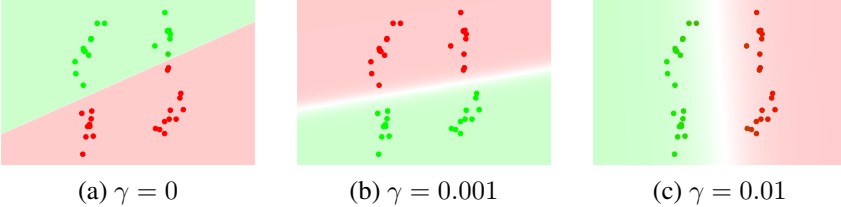

| (a) $\gamma = 0$ | (b) $\gamma = 0.001$ | (c) $\gamma = 0.01$ |

Figure 2: Margin maximization term $\gamma \|\mathbf{v}\|^2$ in our loss (7): low-level clustering results for the softmax linear classifier model (3) with $N = 2$ and different weights $\gamma$. The dots represent data points. The optimal softmax clustering of the data and the decision regions over the whole space are visualized by $\sigma$-weighted color transparency, as in Fig.1(b). The "margin" is a weak-confidence "soft" region around the linear decision boundary lacking color-saturation. For small $\gamma$ the classifier can "squeeze" a narrow-margin linear decision boundary just between the data points, while maintaining arbitrarily hard "decisiveness" on the data points themselves.

## 2 OUR APPROACH TO INFORMATION-BASED CLUSTERING

We are focused on information-based losses for clustering with softmax models that typically enforce "decisiveness" and "fairness". First, In Section 2.1 we argue that common formulations of such losses, e.g. (1) or (5), may produce poor classification margins. We show that some explicit *margin maximization* constraints should be added and motivate regularization of the classifier norm $\|\mathbf{v}\|^2$ for the job, similarly to SVM methods Xu et al. (2004). Section 2.2 introduces our new information-based loss for clustering. First, we motivate a *strong fairness* constraint. Following well-known optimization methodologies Boyd & Vandenberghe (2004), we use *splitting* to divide "decisiveness" and "fairness" into two optimization sub-problems, which are simpler than the original non-convex loss. The overall optimization problem is represented by a Lagrangian function combining network predictions $\sigma$ with auxiliary splitting variables $y$. This Lagrangian is our new self-supervised loss function (10) iteratively solving two simpler sub-problems w.r.t network parameters and splitting variables $y$. While analogous to *pseudo labels* in the previous methods, our loss connects network predictions $\sigma$ and $y$ via the *reverse* cross-entropy $H(\sigma, y)$ making it difficult to call $y$ target or label. Instead, we refer to $y$ as *pseudo confidence*. Section 2.3 derives an efficient EM algorithm solving a sub-problem that estimates $y$.

### 2.1 MARGIN MAXIMIZATION VIA NORM REGULARIZATION

The *average entropy* term in (1), a.k.a. "decisiveness", is recommended in Grandvalet & Bengio (2004) as a general regularization term for semi-supervised problems. They argue that it produces decision boundaries away from all training examples, labeled or not. This seems to suggest larger *classification margins*, which are good for generalization. However, the decisiveness may not automatically imply large margins if the norm of classifier $\mathbf{v}$ in pseudo posterior models (2, 3) is unrestricted, see Figure 2(a). Technically, this follows from the same arguments as in Xu et al. (2004) where regularization of the classifier norm is formally related to the margin maximization in the context of their SVM approach to clustering.

Interestingly, regularization of the norm for all network parameters $[\mathbf{v}, \mathbf{w}]$ is motivated in (4) differently Krause et al. (2010). But, since the classifier parameters $\mathbf{v}$ are included, coincidentally, it also leads to margin maximization. On the other hand, many MI-based methods Bridle et al. (1991); Ghasedi Dizaji et al. (2017); Asano et al. (2020) do not have regularizer $\|\mathbf{v}\|^2$ in their clustering loss, *e.g.* see (5). One may argue that practical implementations of these methods implicitly benefit from the *weight decay*, which is omnipresent in network training. It is also possible that gradient descent may implicitly bound the classifier norm Soudry et al. (2018). In any case, since margin maximization is important for clustering, ideally, it should not be left to chance. Thus, the norm regularization term $\|\mathbf{v}\|^2$ should be explicitly present in any clustering loss for pseudo posterior models.

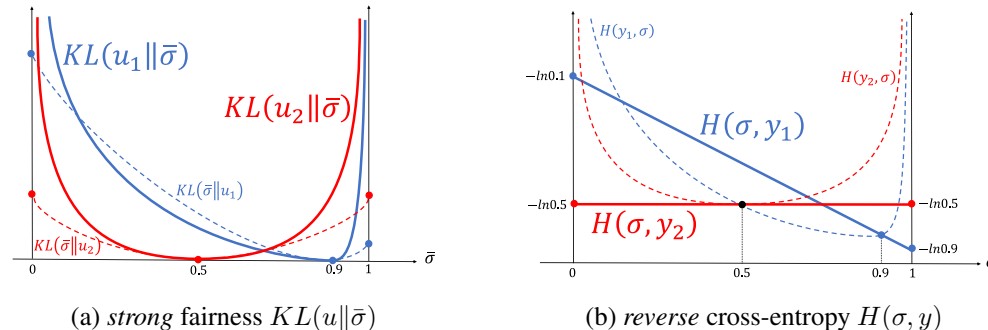

(a) *strong* fairness $KL(u\|\bar{\sigma})$        (b) *reverse* cross-entropy $H(\sigma, y)$

Figure 3: "Forward" vs "reverse": (a) KL-divergence and (b) cross-entropy. Assuming binary classification $K = 2$, we can represent all possible probability distributions as points on the interval [0,1]. The solid curves in (a) illustrate our "strong" fairness constraint, i.e. the *forward* KL-divergence $KL(u\|\bar{\sigma})$ for the average prediction $\bar{\sigma}$. We show two examples of volumetric prior $u_1 = (0.9, 0.1)$ (blue curve) and $u_2 = (0.5, 0.5)$ (red curve). For comparison, the dashed curves represent reverse KL-divergence $KL(\bar{\sigma}\|u)$ commonly used for the fairness in prior art. The solid curves in (b) show our *reverse* cross-entropy $H(\sigma, y)$ w.r.t the network prediction $\sigma$. The dashed curves show the forward cross-entropy $H(y, \sigma)$, which is standard in prior art. The plots in (b) show examples for two fixed estimates of pseudo-confidence $y_1 = (0.9, 0.1)$ (blue curves) and $y_2 = (0.5, 0.5)$ (red curves). The boundedness of $H(\sigma, y)$ represents robustness to errors in $y$. For example, our loss $H(\sigma, y)$ turns-off the training (sets zero-gradients) when the estimated confidence is highly uncertain, see $y_2 = (0.5, 0.5)$ (solid red). In contrast, the standard loss $H(y, \sigma)$ trains the network to copy this uncertainty, e.g observe the optimum $\sigma$ for the dashed curves.

We extend MI loss (1) by combining it with the regularization of the classifier norm $\|\mathbf{v}\|$ encouraging *margin maximization*, as shown in Figure 2

$$
\begin{aligned}
L_{mi+mm} &:= \quad \overline{H(\sigma)} \quad - \quad H(\bar{\sigma}) \quad + \gamma \|\mathbf{v}\|^2 \\
&\stackrel{c}{=} \quad \overline{H(\sigma)} \quad + \quad KL(\bar{\sigma}\,\|\,u) \; + \gamma \|\mathbf{v}\|^2.
\end{aligned}
\tag{7}
$$

We note that Jabi et al. (2021) also extend their information-based loss (6) with the classifier regularization $\|\mathbf{v}\|^2$, but this extra term is used mainly as a technical tool in relating their loss (6) to K-means, as detailed in Section 1.2. They do not discuss its relation to margin maximization.

## 2.2 OUR SELF-SUPERVISED LOSS FUNCTIONS

Below we motivate and put forward new ideas for information-based clustering. First, we observe that the entropy $H(\bar{\sigma})$ in (1) is a weak formulation of the fairness constraint. Indeed, as clear from its equivalent formulation in (7), it is enforced by the *reverse* KL divergence for the average predictions $\bar{\sigma}$. In particular, it assigns a bounded penalty even for a highly unbalanced solutions where $\bar{\sigma}_k = 0$ for some $k$, see the dashed red curve in Fig.3(a). Compare this with the *forward* KL divergence $KL(u\,\|\,\bar{\sigma})$, see the solid red curve. We propose such *zero-avoiding* forward version of KL divergence as a *strong fairness* loss

$$
L_{mi++} \quad := \quad \overline{H(\sigma)} \; + \; \lambda\, KL(u\,\|\,\bar{\sigma}) \; + \; \gamma \|\mathbf{v}\|^2.
\tag{8}
$$

Second, we derive our *self-supervised* loss directly from (8). We use the standard *splitting* technique Boyd & Vandenberghe (2004) to divide optimization of (8) into simpler sub-problems separating the "decisiveness" and "fairness" terms, as follows. Introducing auxiliary *splitting* variables $y \in \Delta^K$, one for each training example $X$, optimization of the loss (8) can be equivalently written as

$$
\min_{\mathbf{v}, \mathbf{w}} \quad \overline{H(\sigma)} \; + \; \gamma \|\mathbf{v}\|^2 \qquad \text{(decisiveness sub-problem)}
$$

$$
\min_{y} \quad KL(u\,\|\,\overline{y}) \qquad \text{(fairness sub-problem)}
$$

$$
s.t. \quad y = \sigma \qquad \text{(consistency constraint)}.
$$

Such constrained optimization problem can be formulated using a Lagrangian function enforcing the equality constraint $y = \sigma$ via the *forward* KL divergence for $y$ (motivated below)

$$L_{our} \quad := \quad \overline{H(\sigma)} \, + \, \beta \, \overline{KL(\sigma \, \| \, y)} \, + \, \lambda \, KL(u \, \| \, \bar{y}) \, + \, \gamma \, \|\mathbf{v}\|^2. \tag{9}$$

The Lagrangian is optimized with respect to both the network parameters and latent variables $y$, but we treat the Lagrange multiplier $\beta$ as a fixed hyper-parameter. Thus, the constraint $y = \sigma$ may not be satisfied exactly and the Lagrangian (9) works only an approximation of the loss (8). Also, one can justify hyper-parameter $\beta = 1$ empirically, see appendix J. Since $\overline{H(\sigma)} + \overline{KL(\sigma \, \| \, y)} = \overline{H(\sigma \, \| \, y)}$, we get the following self-supervised loss formulation

$$L_{our} \quad \overset{\beta = 1}{=} \quad \overline{H(\sigma, y)} \quad + \lambda \, KL(u \, \| \, \bar{y}) \; + \; \gamma \, \|\mathbf{v}\|^2 \tag{10}$$

where the *reverse* cross entropy $\overline{H(\sigma, y)}$ enforces both the decisiveness and consistency $y \approx \sigma$.

There are some notable differences between our loss (10) and existing self-labeling losses. For example, consider the loss (6) proposed in Jabi et al. (2021). Our loss reverses the order of both the KL divergence and the cross-entropy terms. As explained earlier, our version of the KL divergence enforces *stronger fairness*, see Fig.3(a). The reversal of the cross-entropy is motivated in two ways. First, it makes the training of network predictions $\sigma$ robust to errors in noisy estimates $y$, see Figure 3(b), as the pseudo-confidence estimates $y$ are not guaranteed to be accurate. Second, it enforces strong consistency of $y$ with the predictions $\sigma$, which work as target distributions for $y$. Thus, when optimizing with respect to $y$, our loss (10) strongly enforces both the fairness and consistency $y \approx \sigma$. Yet, this well-constrained optimization with respect to $y$ allows an efficient EM solver derived in Section 2.3.

## 2.3 EM ALGORITHM

Expectation-maximization (EM) algorithm Bishop (2006) is commonly used in the context of *generative* methods estimating probability density functions for given data. To estimate parametric density models, their optimal MLE parameters can be computed by maximizing the likelihoods. If data is labeled, estimating separate density functions for each class is relatively straightforward. But if the data is unlabeled, it is common to use EM algorithm to estimate more complex multi-modal densities, e.g. GMMs, that use hidden latent variables (analogous to pseudo-confidence $y$) that are estimated together with the model parameters. Such methods can be used for unsupervised clustering. Unlike our pseudo-confidence estimates $y$, their latent variables do correspond to the true Bayesian posteriors. Their $\arg \max$ is the optimal classifier minimizing expected classification errors Duda et al. (2000); Shalev-Shwartz & Ben-David (2014) assuming that the density functions correctly describe the data. The problem is that complex data requires complex density models that are expensive or impossible to estimate, which motivates discriminative approaches, see Figure 1.

Below we present a new efficient algorithm for optimizing our *discriminative* MI-based loss (10) with respect to the pseudo-confidence estimates $y$ when the model prediction is fixed, *i.e.* $\sigma$ and $\mathbf{v}$. Using the *variational inference* Bishop (2006), we derive a new EM algorithm introducing a different type of latent variables, $K$ distributions $S^k \in \Delta^M$ representing normalized support for each cluster over $M$ data points. We refer to each vector $S^k$ as a *normalized cluster $k$*. Note the difference with distributions represented by pseudo-posteriors $y \in \Delta^K$ showing support for each class at a given data point. Since we explicitly use individual data points below, we will start to carefully index them by $i \in \{1, \ldots, M\}$. Thus, we will use $y_i \in \Delta^K$ and $\sigma_i \in \Delta^K$. Individual components of distribution $S^k \in \Delta^M$ corresponding to data point $i$ will be denoted by scalar $S_i^k$.

First, we expand our loss (10) introducing the latent variables $S^k \in \Delta^M$

$$L_{our} \quad \overset{c}{=} \quad \overline{H(\sigma, y)} + \lambda \, H(u, \bar{y}) + \gamma \, \|\mathbf{v}\|^2 \tag{11}$$

$$= \quad \overline{H(\sigma, y)} - \lambda \sum_k u^k \ln \sum_i S_i^k \frac{y_i^k}{S_i^k M} + \gamma \, \|\mathbf{v}\|^2$$

$$\leq \quad \overline{H(\sigma, y)} - \lambda \sum_k \sum_i u^k S_i^k \ln \frac{y_i^k}{S_i^k M} + \gamma \, \|\mathbf{v}\|^2 \tag{12}$$

Due to the convexity of negative $\log$, we apply the Jensen's inequality to derive an upper bound, i.e. (12), to $L_{our}$. Such bound becomes tight when:

$$\text{E step :} \qquad S_i^k = \frac{y_i^k}{\sum_j y_j^k} \qquad\qquad (13)$$

Then, we fix $S_i^k$ as (13) and solve the Lagrangian of (12) with simplex constraint to update $y$ as:

$$\text{M step :} \qquad y_i^k = \frac{\sigma_i^k + \lambda M u^k S_i^k}{1 + \lambda M \sum_c u^c S_i^c} \qquad\qquad (14)$$

We run these two steps until convergence. Note that the minimum $y$ is guaranteed to be globally optimal since (11) is convex w.r.t. $y$ (Appendix. A). The empirical convergence rate is within 15 steps on MNIST. As for the computational complexity, while there are other optimization methods of the same order $O(NK \cdot Iteration)$, such as Projected Gradient Descent, our EM approach is extremely easy to implement and can be very fast using the highly optimized built-in functions from standard Pytorch library that supports GPU.

Inspired by Springenberg (2015); Hu et al. (2017), we also adapted our EM algorithm to allow for updating $y$ within each batch. In fact, the mini-batch approximation of (11) is an upper bound. Considering the first two terms of (11), we can use Jensen's inequality to get:

$$\overline{H(\sigma, y)} + \lambda\, H(u, \bar{y}) \;\; \leq \;\; \mathbb{E}_B \big[ \overline{H_B(\sigma, y)} + \lambda\, H(u, \bar{y}_B) \big] \qquad\qquad (15)$$

where $B$ is the batch randomly sampled from the whole dataset. Now, we can apply our EM algorithm to update $y$ in each batch, which is even more efficient. Compared to other methods Ghasedi Dizaji et al. (2017); Asano et al. (2020); Jabi et al. (2021) which also use the auxiliary variable $y$, we can efficiently update $y$ on the fly while they only update once or just a few times per epoch due to the inefficiency to update $y$ for the whole dataset per iteration. Interestingly, we found that it is actually important to update $y$ on the fly, which makes convergence faster and improves the performance significantly (Appendix. C). We use such "batch version" EM throughout all the experiments. Our full algorithm for the loss (10) is summarized in the Appendix. B.

## 3 Experimental results

Our experiments start from pure clustering on fixed low-level features to joint clustering with feature learning. We have also compared different losses on weakly-supervised classification. Note that our goal is comparing different losses together with their own optimization algorithms, thus we keeping our experimental setup as simple as possible to reduce the distraction factors for analysis.

### 3.1 Low-level clustering

In this section, we test our loss (10) with a simple linear classifier on MNIST Lecun et al. (1998) by using the (fixed) original features of the images. We compare it to Kmeans and (4). The detailed experimental settings and evaluation methods can be found in Appendix. G and F.

|  | Kmeans | MI Bridle et al. (1991); Krause et al. (2010) | Our |
|---|---|---|---|
| accuracy | 53.2% Hu et al. (2017) | 60.2%(3.7) | **60.8%(1.1)** |

Table 1: Comparison of different losses on MNIST without learning features.

In Table 1, we report the mean accuracy and standard deviation. Note that discriminative clustering methods perform consistently much better than Kmeans ($\geq 7\%$) while our approach achieves a bit higher accuracy but is more robust. Also, a low-level ablation study can be found in Appendix. G.

### 3.2 Deep clustering

In this section, we add deep neural networks for learning features while doing the clustering simultaneously. We use four standard benchmark datasets: STL10 Coates et al. (2011), CIFAR10/CIFAR100 Torralba et al. (2008) and MNIST Lecun et al. (1998). From Table 2, it can

be seen that our approach consistently achieves the best or the most competitive results in terms of accuracy. Experimental settings and more detailed discussion are given in Appendix. H.

| | STL10 | CIFAR10 | CIFAR100-20 | MNIST |
|---|---|---|---|---|
| MI-D$^\star$ Hu et al. (2017) | 25.28%(0.5) | 21.4%(0.5) | 14.39%(0.7) | 92.90%(6.3) |
| IIC$^\star$ Ji et al. (2019) | 24.12%(1.7) | 21.3%(1.4) | 12.58%(0.6) | 82.51%(2.3) |
| SeLa$^\S$ Asano et al. (2020) | 23.99%(0.9) | 24.16%(1.5) | **15.34%(0.3)** | 52.86%(1.9) |
| MI-ADM$^\S$ Jabi et al. (2021) | 17.37%(0.9) | 17.27%(0.6) | 11.02%(0.5) | 17.75%(1.3) |
| Our$^{\star,\S}$ | **25.33%(1.4)** | **24.16%(0.8)** | 15.09%(0.5) | **93.58%(4.8)** |

Table 2: Quantitative results of accuracy for unsupervised clustering methods. We only use the 20 coarse categories for CIFAR100. We reuse the code published by Ji et al. (2019); Asano et al. (2020); Hu et al. (2017) and implemented the optimization for loss of Jabi et al. (2021) according to the paper. $\star$: all variables are updated for each batch. $\S$: loss formula has pseudo-label.

### 3.3 WEAKLY-SUPERVISED CLASSIFICATION

We also test different methods over different levels of (very) weak supervision on STL10. In Table 3, we can see that our approach still shows very competitive results especially with weaker supervision. More details are given in Appendix. I including another test on CIFAR 10.

| | 0.1 | 0.05 | 0.01 |
|---|---|---|---|
| Only seeds | 40.27% | 36.26% | 26.1% |
| + MI-D Hu et al. (2017) | **47.39%** | 40.73% | 26.54% |
| + IIC Ji et al. (2019) | 44.73% | 33.6% | 26.17% |
| + SeLa Asano et al. (2020) | 44.84% | 36.4% | 25.08% |
| + MI-ADM Jabi et al. (2021) | 45.83% | 40.41% | 25.79% |
| + Our | 47.20% | **41.13%** | **26.76%** |

Table 3: Quantitative results for weakly-supervised classification on STL10. 0.1, 0.05 and 0.01 correspond to different ratio of labels used for supervision. "Only seeds" means that we only use standard cross-entropy loss on labelled data for training.

## 4 CONCLUSIONS

We proposed a new self-supervised loss function for information-based discriminative clustering with pseudo-posterior (softmax) models, deep or low-level. Our loss enforces strong *zero-avoiding* fairness and explicitly encourages margin maximization. We show that the latter is achieved by regularizing the norm of the classifier in the network output layer. While this often happens implicitly by omnipresent *weight decay*, it is important to understand the margin maximization effect of the norm regularization. As we show, the "decisiveness" alone is not sufficient for good margins.

In contrast to prior work, we advocate the reverse cross entropy, where network predictions $\sigma$ are swapped with pseudo-confidence estimates $y$. This makes network training robust to errors in $y$. For example, the predictions are not forced to copy uncertain $y$. On the other hand, the predictions $\sigma$ become targets for estimated distributions $y$. Using variational inference, we derive an efficient EM algorithm to optimize our loss with respect to auxiliary variables $y$.

We compare our loss and EM algorithm versus prior losses and the corresponding optimization methods. Our approach consistently produces the top or near-top results on all tested unsupervised and weakly-supervised benchmarks, while the previous methods have high variation in their relative performance.

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

## A  PROOF

**Lemma 1** *Given fixed $\sigma_i \in \Delta^K$ where $i \in \{1, ..., M\}$ and $u \in \Delta^K$, the objective*

$$E(y) = -\frac{\beta}{M} \sum_i \sum_k \sigma_i^k \ln y_i^k - \lambda \sum_k u_k \ln \frac{\sum_i y_i^k}{M}$$

*is convex for $y$, where $y_i \in \Delta^K$.*

*Proof.* First, we rewrite $E(y)$

$$E(y) = -\sum_k \left( \frac{\beta}{M} \sum_i \sigma_i^k \ln y_i^k + \lambda u_k \ln \frac{\sum_i y_i^k}{M} \right)$$

$$:= -\sum_k f_k(y^k) \tag{16}$$

Next, we prove that $f_k : R_{(0,1)}^M \to R$ is concave based on the definition of concavityBoyd & Vandenberghe (2004) for any $k \in \{1, ..., K\}$. Considering $x = (1 - \alpha)x_1 + \alpha x_2$ where $x_1, x_2 \in R_{(0,1)}^M$ and $\alpha \in [0, 1]$, we have

$$f_k(x) = \frac{\beta}{M} \sum_i \sigma_i^k \ln \left( (1 - \alpha)x_{1i} + \alpha x_{2i} \right) + \lambda u_k \ln \frac{\sum_i \left( (1 - \alpha)x_{1i} + \alpha x_{2i} \right)}{M}$$

$$\geq \frac{\beta}{M} \sum_i (1 - \alpha)\sigma_i^k \ln x_{1i} + \alpha \sigma_i^k \ln x_{2i}$$

$$+ \lambda u_k \left( (1 - \alpha) \ln \frac{\sum_i x_{1i}}{M} + \alpha \ln \frac{\sum_i x_{2i}}{M} \right)$$

$$= (1 - \alpha) f_k(x_1) + \alpha f_k(x_2)$$

The inequality uses Jensen's inequality. Now that $f_k$ is proved to be concave, $-f_k$ will be convex. Then $E(y)$ can be easily proved to be convex using the definition of convexity with the similar steps above.

## B  OUR ALGORITHM

---

**Algorithm 1:** Optimization for our loss

---

**Input**   : network parameters $[\mathbf{v}, \mathbf{w}]$ and dataset
**Output:** network parameters $[\mathbf{v}^*, \mathbf{w}^*]$
**for** *each epoch* **do**
  **for** *each iteration* **do**
    Initialize $y$ by the network output at current stage as a warm start;
    **while** *not convergent* **do**
      $S_i^k = \frac{y_i^k}{\sum_j y_j^k}$;
      $y_i^k = \frac{\sigma_i^k + \lambda M u^k S_i^k}{1 + \lambda M \sum_c u^c S_i^c}$;
    **end**
    Update $[\mathbf{w}, \mathbf{v}]$ using loss $\overline{H_B(\sigma, y)} + \gamma \|\mathbf{v}\|^2$ via stochastic gradient descent
  **end**
**end**

---

## C  LOSS CURVE

## D  NETWORK ARCHITECTURE

The network structure is VGG-style and adapted from Ji et al. (2019).

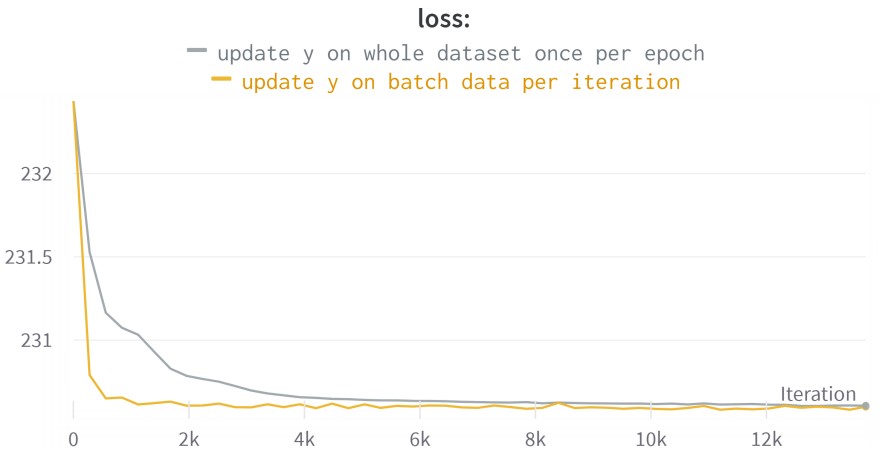

Figure 4: Loss (10) curves for different update setting on $y$. This is generated with just a linear classifier on MNIST. We use the same initialization and run both for 50 epochs. The gray line has an accuracy of 52.35% while the yellow one achieves 63%.

| Grey(28x28x1) | RGB(32x32x3) | RGB(96x96x3) |
|---|---|---|
| 1xConv(5x5,s=1,p=2)@64
1xMaxPool(2x2,s=2)
1xConv(5x5,s=1,p=2)@128
1xMaxPool(2x2,s=2)
1xConv(5x5,s=1,p=2)@256
1xMaxPool(2x2,s=2)
1xConv(5x5,s=1,p=2)@512 | 1xConv(5x5,s=1,p=2)@32
1xMaxPool(2x2,s=2)
1xConv(5x5,s=1,p=2)@64
1xMaxPool(2x2,s=2)
1xConv(5x5,s=1,p=2)@128
1xMaxPool(2x2,s=2)
1xConv(5x5,s=1,p=2)@256 | 1xConv(5x5,s=2,p=2)@128
1xMaxPool(2x2,s=2)
1xConv(5x5,s=2,p=2)@256
1xMaxPool(2x2,s=2)
1xConv(5x5,s=2,p=2)@512
1xMaxPool(2x2,s=2)
1xConv(5x5,s=2,p=2)@1024 |
| 1xLinear(512x3x3,K) | 1xLinear(256x4x4,K) | 1xLinear(1024x1x1,K) |

Table 4: Network architecture summary. s: stride; p: padding; K: number of clusters. The first column is used on MNIST Lecun et al. (1998); the second one is used on CIFAR10/100 Torralba et al. (2008); the third one is used on STL10 Coates et al. (2011). Batch normalization is also applied after each Conv layer. ReLu is adopted for non-linear activation function.

## E   DATASET SUMMARY

| STL10 | CIFAR10 | CIFAR100-20 | MNIST |
|---|---|---|---|
| 13000 | 60000 | 60000 | 70000 |
| 96x96x3 | 32x32x3 | 32x32x3 | 28x28x1 |

Table 5: Dataset summary for unsupervised clustering.

Table 5 indicates the number of (training) data and the input size of each image for the unsupervised clustering. Training and test sets are the same.

As for **weakly-supervised** classification on STL10, we use 5000 images for training and 8000 images for testing. We only keep a certain percentage of ground-truth labels for each class of training data. The accuracy is calculated on test set by comparing the hard-max of prediction to the ground-truth.

## F EXPERIMENTAL EVALUATION

As for the evaluation on unsupervised clustering, we set the number of clusters to the number of ground-truth categories and we adopt the standard method Kuhn (1955) by finding the best one-to-one mapping between clusters and labels. We use the accuracy as the measure for both unsupervised and weakly-supervised settings while the latter calculates the accuracy on the test set.

## G LOW-LEVEL CLUSTERING

As for the experiments on MNIST Lecun et al. (1998), we transform the original image values linearly into $[-1, 1]$ and use the flattened images as input features. Note that here we only use a linear classifier without training any features. We employ stochastic gradient descent with learning rate $0.07$ to update $\mathbf{v}$ in (4) and (10). We use the same (random) intialization for both losses and run each 6 times up to 50 epochs per run. We use 250 for batch size. We set $\gamma = 0.01$ for both and use $\lambda = 100$ for (10) and $\lambda = 1.3$ for (4).

**We fix the hyperparameter values for** (9) **and** (4) **throughout the whole experimental sections**.

We also conducted an ablation study on toy examples as shown in Figure. 5. We use the normalized X-Y coordinates of the data points as the input. We can see that each part of our loss is necessary for obtaining a good result. Note that, in Figure 5 (a), (c) of 3-label case, the clusters formed are the same, but the decision boundaries which implies the generalization are different. This emphasizes the importance of including $L2$ norm of $\mathbf{v}$ to enforce maximum margin for better generalization.

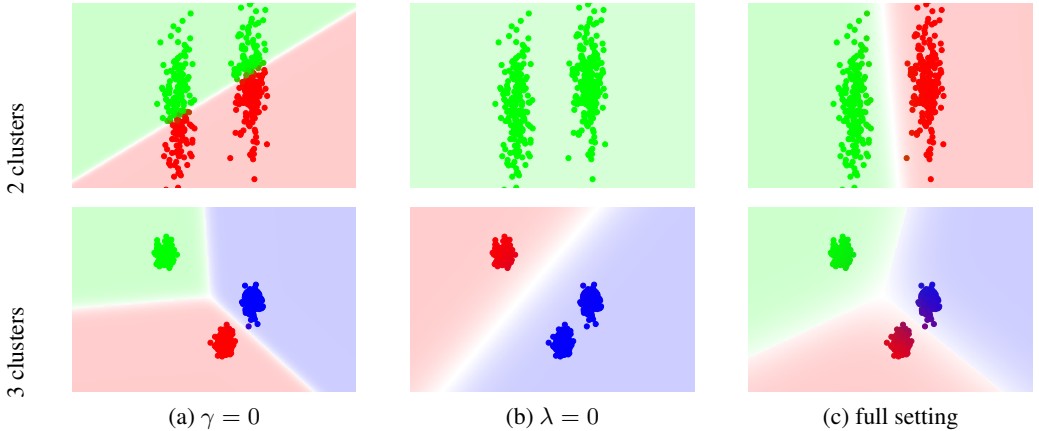

Figure 5: "Shallow" ablation study on toy examples.

## H DEEP CLUSTERING

We add deep neural networks for learning features while doing the clustering simultaneously. We use four standard benchmark datasets: STL10 Coates et al. (2011), CIFAR10/CIFAR100 Torralba et al. (2008) and MNIST Lecun et al. (1998). As for the architectures, we followed Ji et al. (2019) to use VGG11-like network structures whereas we use it for both gray-scale and RGB images with some adjustments as shown in Appendix. D.

Besides the clustering loss, we also adopted self-augmentation techniques, following Hu et al. (2017); Ji et al. (2019); Asano et al. (2020). Such technique is important for enforcing neural networks to learn augmentation-invariant features, which are often semantically meaningful. While Ji et al. (2019) designed their loss directly based on such technique, our loss and Krause et al. (2010); Asano et al. (2020); Jabi et al. (2021) are more general for clustering without any guarantee to generate semantic clusters. Thus, for fair comparison and more reasonable results, we combine such augmentation technique into network training. Specifically, we achieved this by set-

ting $\sigma_i = \mathbb{E}_t[\sigma(\mathbf{v}^\top f_\mathbf{w}(t(X_i)))]$. For each image, we generate two augmentations sampled from "horizontal flip", "rotation" and "color distortion".

We use Adam Kingma & Ba (2015) with learning rate $1e^{-4}$ for optimizing the network parameters. We set batch size to 250 for CIFAR10, CIFAR100 and MNIST, and we use 160 for STL10. In Table 2, we report the mean accuracy and Std from 6 runs with different initializations while we use the same initialization for all methods in each run. We still use 50 epochs for each run and all methods reach convergence within 50 epochs.

As for other methods in Table 2, MI-D has the most comparable results to us, in part because our loss can be seen as an approximation to the MI and we both update all variables per batch. SeLa achieves relatively better results on other three datasets than MNIST, because it enforces a *hard* constraint on "fairness" and MNIST is the only one out of four sets that is not exactly balanced. In real world, the data we collect is mostly not exactly balanced. This could be the reason why such method is better for the unsupervised representation learning where over-clustering can be employed and real clusters become less important. MI-ADM only updates the pseudo-labels once per epoch, thus easily leading the network towards a trap of local minimum created by the incorrect pseudo-labels through the forward cross-entropy loss as illustrated in Figure 3.

## I  WEAKLY-SUPERVISED CLUSTERING

We use the same experimental settings as that in unsupervised clustering except for two points: 1. We add cross-entropy loss on labelled data; 2. We separate the training data from test data while we use all the data for training and test in unsupervised clustering.

While MI-ADM is the worst according to Table 2, it is improved significantly in weakly-supervised setting. This might be a sign that the advantage of more frequent update on variables in unsupervised clustering is waning since the seeds help the network keeping away from some bad local minima.

Below is the result on CIFAR 10.

|            | 0.1     | 0.05    | 0.01    |
|------------|---------|---------|---------|
| Only seeds | 58.77%  | 54.27%  | 39.01%  |
| + MI-D     | 65.54%  | 61.4%   | 46.97%  |
| + IIC      | **66.5%** | 61.17%  | 47.21%  |
| + SeLa     | 61.5%   | 58.35%  | 47.19%  |
| + MI-ADM   | 62.51%  | 57.05%  | 45.91%  |
| + Our      | 66.17%  | **61.59%** | **47.22%** |

## J  HYPERPARAMETER $\beta$

Below is an empirical justification for setting hyper-parameter $\beta = 1$ in the loss (9). The first two terms in (9) can be written as $(1 - \beta)\overline{H(\sigma)} + \beta H(\sigma, y)$. If $\beta > 1$ then the negative entropy pushes predictions $\sigma$ away from one-hot solutions weakening the decisiveness. On the other hand, if $\beta < 1$ then the loss is non-convex w.r.t $\sigma$ that may trap gradient descent in bad local minima, as illustrated by the plots for $y = (0.9, 0.1)$ in Figure 6

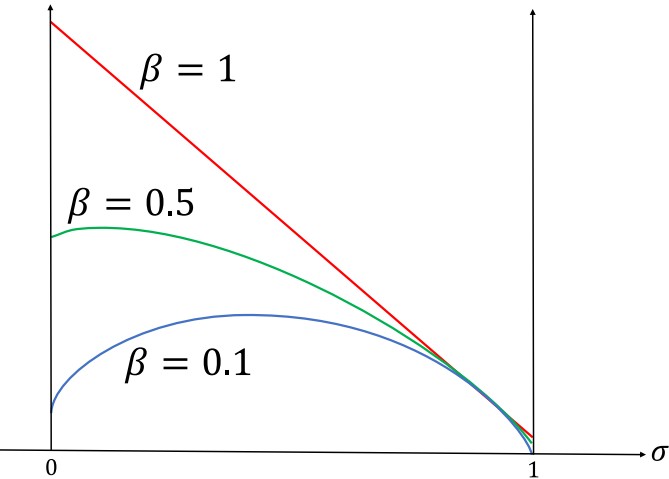

Figure 6: $(1 - \beta)H(\sigma) + \beta H(\sigma, y)$

