# OpenReview forum: "Revisiting Information-Based Clustering with Pseudo-Posterior Models"
_ICLR.cc/2023/Conference — Submitted to ICLR 2023_

### Official Review · Reviewer_oWJS · 2022-10-24

**Confidence:** 5
**Correctness:** 1
**Technical Novelty And Significance:** 1
**Empirical Novelty And Significance:** 1
**Recommendation:** 1

**Clarity, Quality, Novelty And Reproducibility:**

The writing and organization of this paper are poor. It is hard to read and follow the authors’ idea. The proposed normalization term is not novel. The reproducibility might be good since the proposed method is quite trivial.

**Strength And Weaknesses:**

- Weakness:
    1. The organization of the paper is poor. For example, the authors spend about two pages discussing previously well-known methods and the corresponding properties. It is not clear what problem the authors want to tackle.
    2. The current “contributions” (especially the second) are not exactly contributions. They are more like details in the proposed method.
    3. Fig. 2 indicates that the weight parameter needs to be carefully tuned to make the normalization term work.
    4. Experiments on MNIST show almost the same performance for the proposed method and previous MI-based ones. Moreover, it is unfair to conduct K-means on the raw data while using a 1-layer classifier for the proposed method.
    5. The experimental results on deep clustering benchmarks are not correct. For example, the results for IIC do not match those reported in its original paper. Thus I doubt the correctness of the conducted experiments.
    6. The authors claim the normalization term is helpful for those pseudo label based clustering methods. But no corresponding experiments are conducted to support the claim.
    7. Parameter analysis on the weight gamma is missing.
    8. Minor: It is really strange to name the loss L_our.

**Summary Of The Paper:**

This paper proposes adding a normalization term to the existing MI-based clustering objective.

**Summary Of The Review:**

As mentioned above, this paper is of bad quality in organization and writing. The proposed method is neither novel or effective. The experiments are also not conducted correctly and sufficiently.

---

### Official Review · Reviewer_7XGP · 2022-10-24

**Confidence:** 2
**Correctness:** 4
**Technical Novelty And Significance:** 2
**Empirical Novelty And Significance:** 3
**Recommendation:** 6

**Clarity, Quality, Novelty And Reproducibility:**

The work is well-written, although I think there could be more focus on the justification of the proposed algorithm, as opposed to the background.

**Strength And Weaknesses:**

The authors provided intuitions on the proposed modifications, which are based on past interpretations for feature-space clustering.  The proposed algorithm appears to have competitive performance, but I am not familiar with this literature and cannot judge on the significance of the empirical results.

On the downside, the novelty is a bit lacking: the use of $\ell_2$ regularization is ubiquitous, and its margin-promoting behavior is known in different but related settings (e.g., softmax classification); the replacement of reverse KL with forward KL is straightforward given the fairness interpretation of the former.  The interpretation of MI as enforcing "fairness" and "decisiveness" is also somewhat arbitrary and simplistic, as they appear to provide no further concrete insights, than the proposed algorithm.  Thus, it appears that the significance of the work hinges on the empirical side.


**Summary Of The Paper:**

This work observed a connection between mutual information-based clustering loss and margin-maximization, and proposed two changes to the process: (i) explicit $\ell_2$ regularization to promote the margin maximization behavior, and (ii) replacing the entropy term in MI, which can be viewed as a reverse KL divergence, with a forward KL divergence.  The modifications lead to improved performance on weakly supervised classification tasks.

**Summary Of The Review:**

+ The authors made some observations on the MI loss for clustering, which leads to an algorithm that appears to perform well

- The motivating discussions are somewhat simplistic, and not fully convincing

---

### Official Review · Reviewer_3fzk · 2022-10-24

**Confidence:** 4
**Correctness:** 3
**Technical Novelty And Significance:** 2
**Empirical Novelty And Significance:** 2
**Recommendation:** 3

**Clarity, Quality, Novelty And Reproducibility:**

The text is a bit confusing which makes it a bit difficult to realize the contributions.

Novelty: reversing the Cross-Entropy or the KL are interesting ideas, that are worth talking about.

Reproducibility: I wouldn't be able to implement the model. The theory is fine but it does help with the implementation. What is y? is it just an ad-hoc variable? is it part of the network? what is the difference between $\sigma$ and y in the implementation?

**Strength And Weaknesses:**

+ Reversing the cross-entropy for the pseudo-labels is indeed a good idea. In my understanding, it can replace pruning or selecting the pseudo-labels since the low-confident ones will "naturally"  have little influence on the training.
+ Inverting the KL is also interesting but I don't really see a huge benefit. Fair enough.

- The writing is confusing. Arguments cross each other, the construction is not linear and some variables are not clearly defined.
- The experimental section is poor. Several Ablation analyses are missing.


**Summary Of The Paper:**

The paper discusses information theory-based clustering and self-supervised learning.
The main contribution lies in the use of "reversed" KL divergence and cross-entropy.


The paper is a bit confusing, so I present here my understanding of the theory.

Clustering can be done by maximizing the Mutual Information over the class prediction, $\sigma$:
$$\min_\sigma -MI(\sigma, X)$$
According to Bridle et al. (1991) the latter can be approximated as follows, where the $avg$ means average:
$$\min_\sigma avg[H(\sigma)] - H(avg({\sigma}))$$
The last term can be rewritten as a KL divergence wrt to a uniform distribution plus a constant that we omit:
$$\min_\sigma avg[H(\sigma)] + KL(avg(\sigma)|| Unif)$$
The clustering is done in a feature space learned by a network with weight w, and the clustering is controlled by one/two vectors v.
Krause et al. 2010  suggested an l2-regularization of all the weights involved that can be weighted.
$$\min_\sigma avg[H(\sigma)] + KL(avg(\sigma)|| Unif) + \gamma ||v,w||^2$$
The idea is that controlling the norm of v (classifier) also controls the margin. exactly like an SVM.
Controlling the norm of w helps the training.

*First contribution*: the authors reverse the KL (Eq10) and weigh it (the authors omit w).
$$\min_\sigma avg[H(\sigma)] + \lambda KL(Unif||avg(\sigma)) + \gamma ||v,w||^2$$
The argument is to prevent $avg(\sigma)$ from being a one-hot vector in a more strict manner than if the KL is computed the other way around. $avg(\sigma)$ being a one-hot vector ( or at least that some of its dimensions go to zero) means that some clusters collapse aka are empty.

Now let's assume we have some (soft) pseudo-labels $y$. If we step back and state what we try to achieve:
1/ Good class separation with a margin: $\min_{v,w,\sigma} avg[H(\sigma)] + \gamma||v,w||^2$
2/ All the clusters should be used (fairness): $KL(Unif||avg(y))$
3/ Consistency: the pseudo labels match the class assignments: $y = \sigma$

*Second contribution*: Using a KL to match $\sigma$ and $y$.
We invoke Lagrange to merge them all:
$$ L = avg[H(\sigma)] + \beta \cdot avg[KL(\sigma||y)] + \lambda KL(Unif||avg(y)) + \gamma||v,w||^2$$If we set $\beta=1$, it simplifies to the final loss:
$$ L = avg[H(\sigma,y)] + \lambda KL(Unif||avg(y)) + \gamma||v,w||^2$$
where the cross entropy between class prediction $\sigma$ and pseudo-label $y$ is reverse compared to usual.
For $y$ fixed, $H(\sigma,y)$ is a line that is horizontal if the model is not confident on the pseudo-label $y$. This means the gradient becomes close to zero. So no need to prune the pseudo-labels!

**Summary Of The Review:**

The paper brings interesting ideas.
After reading it all, I am still unsure what is $y$. how does it differ from $\sigma$ and how do we compute all of these?

The authors try to build a discussion about clustering with information theory, but it turns confusing. There are too many arguments not very well organized. They compare themselves too much to Jabi et al. to the point that the main contributions are a bit hidden. I think the whole presentation could be greatly simplified.

I do find the ideas interesting, but the paper/presentation is too confusing.

If I understood well, you're doing a centroid based clustering (hence the v) where you want the separation to be controlled with information theory tools. It is a fair endeavor. But then you mix geometry (l2 regularization) and information theory (entropy).
Can't you control the margin using only pure entropy-based tools?

Regarding clustering with information theory, I would suggest the authors have a look at:
https://arxiv.org/abs/1406.1222
In this paper, they use pure information theory tools and achieve pretty impressive results.





Figure 1 is a bit unfair. depending on the initialization, k-means or GMM can split this dataset correctly.
Also for a softer boundary, then just use GMM which would be fairer than k-means, since the latter uses hard assignments while the proposed method relies on soft assignments, like GMM.

Eq 5: I think you want to say that y arises from a uniform distribution. The correct notation is $\bar{y] \tilde u$.

page4 The topic moves from clustering to self-supervised learning without real warning.

Fig3: The caption is confusing. So are the plots. Regarding Fig 3.a: why allowing $\sigma$ to have zero coordinates not acceptable? You might also want to let the network decide and close some clusters.

Page 6 optimization problem. Why is $\sigma$ not under the first min? I put it in my summary. Maybe I am wrong. Maybe sigma is a function of v and w. it's not clear.
Why do you use y in the second minimization, while it was not in the previous equations? I am not sure this helps to understand how the equations connect.

Section 3.1: Why Equation 4 and not 10? I thought Eq10 was your final model.
Also, k-means can cluster raw MNIST with more than 53% accuracy, using sklearn implementation.

Equation 10 relies on pseudo labels. It is thus also dependent on the initialization. So I would be interested to see how it works when not supported by a network.

Appendix G: Why Equation 9 and not 10? There are no gamma and lambda in equation 4. Figure 5, which one is Eq4 which one is Eq.9?

Table 2: For IIC, did you use over-clustering heads as it is recommended in the paper?
The discussion in Appendix H should be in the main text.

---

### Author Response · Authors · 2022-11-18
**Big Picture + Importance of Figure 1 (please read analysis on p.2)**

Before responding to some specific comments below, we summarize three main reasons why this work must be (and will be) published, at ICLR or not. First of all, our paper disproves the main theoretical claim (in the title) of a recent TPAMI paper [Jabi et al 2021] wrongly stating the equivalence between the standard K-means loss and entropy-based clustering losses, e.g. (6). TPAMI is the most influential ML journal and our correction is urgent. In fact, reviewer 3fzk refers to this as a ``mix of geometry and information theory'', but mistakenly attributes this confusion in [Jabi et al 2021] to our work. Our Figure 1 provides a simple counterexample to the claim, but we also point out
specific technical errors in their proof (see our footnotes 2,3).

Second, Figure 1 is also meant to motivate our readers to learn about the standard entropy-based clustering criteria for (ubiquitous nowadays) posterior/softmax models [MacKay et al 1991, Perona et al 2010]. This general clustering methodology is undeservedly little-known to the broader ML community for two reasons: (A) it was previously presented only in the context of complex (non-linear, deep) softmax models obfuscating the basics and (B) because there is confusion even among the researchers who know about it
(see the first paragraph). One of our goals is to explain/clarify this general form of discriminative clustering to the community. Simple Figure 1 is central to this.

Last, but not least, we propose and motivate several technical novelties for entropy-based self-supervised clustering losses: (A) we show significant limitations of the standard Shannon's cross-entropy $H(y,\sigma)$ in case of soft targets $y$ and motivate an alternative formulation, the reverse cross-entropy that is robust to label uncertainty, see Fig.3b and our new Table 1 at the bottom of this rebuttal. (B) we show the importance of the classifier norm regularization for margin maximization, it is missing in the related prior art. (C) we design a new EM algorithm with closed-form EM steps (13,14). This makes our loss almost as easy-to-use as K-means. In part, our MI-loss variant is motivated by its amenability to an efficient EM solver.

$\text{\bf [oWJS] unfair to compare K-means on raw data while using a 1-layer classifier (Fig 1)}$ - why? both models are linear, have the same number of parameters, and are applied to the same raw data. In fact, K-means can be interpreted as a self-supervised variant of the linear Gaussian classifier with $\Sigma_k=I$.

$\text {\bf [3fzk] depending on initialization, K-means or GMM can work (Fig.1)}$ - we show global optima for K-means. GMM will work but it is more complex. This is fully discussed on p.2 under Fig.1.

$\text{\bf [3fzk] what is $y$}$ - these are auxiliary/latent optimization variables in prior work (Sec.1.1 above (5,6)) and in our work (Sec 2.2 below (8)). Such pseudo-labels are jointly estimated with the model parameters. This is analogous to K-means jointly optimizing cluster assignments (data pseudo-labels) and cluster means (model parameters), but not equivalent, despite [Jabi 2021].

$\text{\bf [7XGP] fairness and decisiveness interpretation of MI is simplistic}$ - this opinion applies to [MacKay et al 1991, Perona et al 2010] and all of their follow-up, including well-cited ICLR papers [Vedaldi et al 2020]. Partially agreeing, we call (1) pseudo-MI on p.1. But our discussion of fairness and decisiveness, e.g. Fig.1, may convince the reviewer that such discriminative losses could be more relevant than simplistic l2 errors minimized by K-means and as easy to optimize (our EM).

$\text{\bf [7XGP] replacement of reverse with forward KL is straightforward}$ - simplicity does not diminish the novelty of our conceptually well-motivated "strong fairness". Moreover, only this particular version allows an efficient EM algorithm with closed-form steps.

$\text{\bf [7XGP] margin-promoting by l2 regularization is known in different settings}$ - we cite some, but will gladly cite more references, please provide specific ones.

$\text{\bf [oWJS] IIC results do not match...}$ - we directly compare the loss functions and their training effect in fair experiments with identical backbones and excluding heuristics, which can be added to any method but obfuscate the loss properties. Our clean comparison is easy to reproduce.

$\text{\bf TABLE 1: robustness to uncertain (soft) labels } y$
| $\eta$  | 0  | 0.2  | 0.4  | 0.6 | 0.8 |
| :---: |---|---|---|---|---|
| $H(y,\sigma)$ |  61.22 | 59.09 | 56.17 | 48.08 | 16.98|
| $H(\sigma,y)$ | undef | 60.99 | 59.88 | 58.85 | 40.32 |

Table 1: Standard vs reverse cross-entropy for training with noisy ground truth. We use VGG-4 backbone on corrupted STL10 where $\eta$ is the percentage of ground truth labels replaced by random labels. The "smoothed" targets [Hinton et al 2019] are $y=\eta u + (1-\eta) \\text{ one-hot} (\tilde{y}) $ with uniform distribution $u$ and observed noisy label $\tilde{y}$.

---

### Decision · Program_Chairs · 2023-01-20

**Decision:**

Reject

**Justification For Why Not Higher Score:**

The paper is hard to follow and the experimental section is not reproducible

**Justification For Why Not Lower Score:**

N/A

**Metareview: Summary, Strengths And Weaknesses:**

In the paper, the authors present a new loss function for clustering. They rely on the reverse KL divergence and margin maximization to obtain more robust and fair clustering. The algorithm sounds interesting, but the paper is hard to follow, as pointed out by Reviewers oWJS and 3fzk. Reviewer 3fzk indicated that the proposed loss function is novel and can provide interesting results. The experimental section is too short, and it is hard to reproduce the results shown in the paper. The authors only report a pure clustering result in Section 3.1 (6 lines of text and a table). The other two sections are for classification results that are trained together with the clustering mechanism. The paper needs substantial work before it can be presented at a mayor ML conference.

**Summary Of Ac-Reviewer Meeting:**

N/A